# Organic Photodetectors with Extended Spectral Response Range Assisted by Plasmonic Hot-Electron Injection

**DOI:** 10.3390/nano12173084

**Published:** 2022-09-05

**Authors:** Aiping Zhai, Chenjie Zhao, Deng Pan, Shilei Zhu, Wenyan Wang, Ting Ji, Guohui Li, Rong Wen, Ye Zhang, Yuying Hao, Yanxia Cui

**Affiliations:** 1College of Physics and Optoelectronics, Taiyuan University of Technology, Taiyuan 030024, China; 2Aluminum-Magnesium Based New Material R&D Co., Ltd., Subsidiary of Xing Xian County Economic and Technological Development Zone, Lvliang 035300, China

**Keywords:** organic photodetector, nanostructure, plasmonic resonance, hot electron, near infrared

## Abstract

Organic photodetectors (OPDs) have aroused intensive attention for signal detection in industrial and scientific applications due to their advantages including low cost, mechanical flexibility, and large-area fabrication. As one of the most common organic light-emitting materials, 8-hydroxyquinolinato aluminum (Alq_3_) has an absorption wavelength edge of 460 nm. Here, through the introduction of Ag nanoparticles (Ag NPs), the spectral response range of the Alq_3_-based OPD was successfully extended to the near-infrared range. It was found that introducing Ag NPs can induce rich plasmonic resonances, generating plenty of hot electrons, which could be injected into Alq_3_ and then be collected. Moreover, as a by-product of introducing Ag NPs, the dark current was suppressed by around two orders of magnitude by forming a Schottky junction on the cathode side. These two effects in combination produced photoelectric signals with significant contrasts at wavelengths beyond the Alq_3_ absorption band. It was found that the OPD with Ag NPs can stably generate electric signals under illumination by pulsed 850 nm LED, while the output of the reference device included no signal. Our work contributes to the development of low-cost, broadband OPDs for applications in flexible electronics, bio-imaging sensors, etc.

## 1. Introduction

Photodetectors have been widely used for signal detection in industrial and scientific applications, including environmental monitoring, chemical/biological sensing, day/night surveillance, remote control, and so forth [1,2,3,4,5]. At present, most of commercial photodetectors are based on inorganic semiconductors, such as Si, Ge, and III-V compounds [6,7], which exhibit high sensitivity, low dark current, and high stability. However, inorganic semiconductors are accompanied by demerits, such as poor mechanical flexibility, complex preparation process, high cost, and limited choice of materials. As alternatives, organic semiconductors offer various desirable properties, such as good mechanical flexibility, solution processability, low cost, and rich material choices, which have contributed to the development of high-performance organic photodetectors (OPDs) in the past few years [8,9,10,11,12,13,14]. For example, C. Fuentes-Hernandez et al. proposed a large-area flexible organic photodiode using P3HT: ICBA as the photoactive layer, which could detect very weak light due to its ultra-low noise current of 37 fA [15]. M. Biele et al. reported another organic photodiode based on the Lisicon PV-D4650 polymer, which manifested a linear dynamic range (LDR) as large as 160 dB [16]. F. Guo et al. demonstrated a solution-processed ultraviolet photodetector composed of ZnO nanoparticles blended with PVK [17], of which the external quantum efficiency (EQE) was over 10^5^%, enabled by the trap-assisted carrier tunneling injection effect. Apart from the above-mentioned two-terminal organic photoelectric devices, three-terminal organic phototransistors have also attracted significant attention because they integrate the functions of light detection and signal amplification [18,19].

It is well known that traditional OPDs utilize the organic semiconductors to absorb light, so their operating spectral ranges are determined by the energy bandgaps of typical organic materials. In order to extend the response to infrared range, most efforts have been focused on developing narrow-bandgap organic materials [20,21,22]. However, it is difficult to reduce the bandgap beyond 1500 nm due to the intrinsic molecular structures of organic materials, as this might cause high non-radiative recombination and inefficient exciton dissociation, corresponding to deteriorated dark current and photocurrent.

Notably, plasmonic nanostructures which can yield near-unity absorption at the resonant wavelengths have been successfully introduced into photodetectors made of inorganic semiconductors, e.g., silicon [23,24,25], titanium dioxide [26,27], and zinc oxide [28,29,30], for extending their operating spectral ranges. In those photoelectric devices, light absorption occurs in metals by the excitation of plasmonic resonances, followed by the generation of free carriers in neighboring semiconductors, based on the processes of hot-carrier transfer, charge-transfer transition, or resonant energy transfer [31,32,33]. Among these processes, the plasmonic induced hot-carrier transfer process (PHCT) was the most common, in which the photo-excited carriers with sufficiently high energy in metals can jump over the metal/semiconductor Schottky barrier and then enter into the adjacent semiconductors, which consequently generates photocurrents in the circuit.

It is noted that the PHCT effect on organic–metal interfaces has been studied for decades, but has not attracted great attention because of poor light absorption by planar metal films. In recent years, a few endeavors have been made for demonstrating broadband photodetectors made of organic wide-bandgap materials by inducing hot carriers utilizing complicated metallic structures, which displayed superiority on light absorption, with respect to their planar counterparts. For instance, in 2016, a silver transparent electrode carved with nanohole arrays was introduced into an OPD made of spiro-TTB with energy gap of 3.3 eV [34]. Ascribed to the hot hole transfer from silver into spiro-TTB, the nanohole OPD device could respond to near-infrared light towards 1300 nm. In 2021, Jin et al. also demonstrated a hot-electron OPD [35] comprised of planar multilayer films ITO (Indium Tin Oxide)/Pt/HAT-CN/Al, which could produce remarkable photocurrent responses over a broadband infrared range from 900 nm to 1600 nm; here, HAT-CN is a wide-bandgap organic material. Moreover, the resonant energy transfer induced by plasmonic gold nanoparticles was also researched for enhancing charge-transfer absorption at the NPB/C_60_ interfaces by Peng’s group [36,37], so that near-infrared photodetection could be realized. In their device, a single NPB or C_60_ layer has no absorption in the wavelength range longer than 650 nm.

In this work, we proposed a broadband PHCT-based OPD employing the small-molecule organic semiconductor of 8-hydroxyquinolinato aluminum (Alq_3_). It is well known that Alq_3_ is often used as an organic light-emitting material as well as an electron-injection material, and it can only absorb light with a wavelength shorter than 460 nm. Incorporating plasmonic nanostructures allows the Alq_3_-based photodetector to possess an extended spectral range, offering an alternative strategy of developing low-cost infrared OPD. It is well known that noble metal nanoparticles, such as those made of Au and Ag, were always used as light harvesters to broaden the optical absorption and enhance the light scattering intensity [38,39,40,41]. Compared to Au, Ag has relatively lower resistivity, and it is more inexpensive. Herein, the broadband PHCT-based OPD has a configuration of ITO/Alq_3_/AgNP/Al, in which a thermally evaporated Ag nanoparticles (NPs) layer was decorated in between the Alq_3_-active layer and the Al cathode layer for exciting plasmonic resonances. Ascribed to the excitation of plasmonic resonances, the absorption of light for the ITO/Alq_3_/AgNP/Al device is apparently enhanced over a broadband wavelength range, compared with that of the control device (ITO/Alq_3_/Al). Moreover, with the help of Ag NPs, the initial Ohmic contact at the Alq_3_/Al interface is transformed into the Schottky contact, resulting in a significant suppression of dark current by around two orders of magnitude. Because the amount of the PHCT-induced electron generation in Alq_3_ is at a low level, the inhibition of dark current enables the photoelectric signals to be apparently distinct from the dark currents under a non-zero bias. Consequently, the OPD with Ag NPs can exhibit stable transient photocurrent responses to the pulsed 660 nm and 850 nm LEDs; instead, the control device can hardly give any responses at the two wavelengths. The OPD proposed in this work uses low-cost materials which can be easily processed on either rigid or flexible substrates. Our work contributes to the development of low-cost, large-area photodetectors, along with image sensors having the potential to be applied in wearable electronics, industry visions, etc.

## 2. Materials and Methods

### 2.1. Fabrication of Device

Material information is detailed in Appendix A. The entire fabrication process was performed in a vacuum evaporation chamber. First, the ITO-coated glass was cleaned successively with deionized water, acetone, and isopropyl alcohol for 15 min in each round. It was then transferred into the glove box. Subsequently, an Alq_3_ layer with a thickness of 80 nm was deposited onto the ITO surface with an evaporation rate of 1 Ås^−1^ at a temperature higher than 150 °C. Successively, the Ag NP layer and the Al cathode layer were deposited with the evaporation rates of 0.2 Ås^−1^ and 4 Ås^−1^, until their nominal thicknesses reached 7.5 nm and 10 nm, respectively. Optimization of the Ag NP size is detailed in Appendix A. The evaporation pressure was less than 5 × 10^−4^ Pa. Finally, the plasmonic OPD with the configuration of ITO/Alq_3_/AgNP/Al was cooled down for 30 min in a vacuum before characterizations. All devices have an effective area of 0.04 cm^2^. Here, the control device without Ag NPs was also fabricated for comparison, with a configuration of ITO/Alq_3_/Al. Schematic diagrams of the plasmonic OPD along with the control in the cross-sectional view are shown in Figure 1a,b, respectively. Considering that the island-like three-dimensional Ag clusters are formed in the process of Ag films deposition, the Ag NPs are drawn scattered. The insets include the molecular structure of Alq_3_ and the three-dimensional schematic diagram of the Ag NP layer.

### 2.2. Characterization

Scanning electron microscopy (SEM) images of the surfaces of devices and films were taken by a field emission scanning electron microscope (JSM-7100F, Japan Electron Optics Laboratory, Tokyo, Japan). A spectrometer with an integrating sphere was used to measure the absorption of light at the wavelength range of 300~900 nm. Current–voltage (I−V) characteristics and transient photoresponses of the OPDs at different wavelengths (375 nm, 565 nm, 660 nm, and 850 nm, LEDs from Thorlabs Inc. Newton, New Jersey, USA) were tested using the semiconductor parameter analyzer (B1500A, Agilent, CA, USA). Light intensities were calibrated by a power meter (NOVA II, Ophir, Israel). For the I−V measurements, all samples were placed on a probe station equipped with a microscope in a shield box (Simple PS Probe Station, KeyFactor, Guangdong, China).

## 3. Results and Discussion

Firstly, the influences of Ag NP incorporation on the photodetection performances of the Alq_3_-based OPD are characterized and discussed. The thin curves in Figure 2a,b show the logarithmic I−V characteristics of the control OPD and the plasmonic OPD, respectively, in the dark, with the corresponding linear I−V plots also presented in Appendix A. By comparison, it is apparent that under forward bias above 2 V, the control OPD has a high dark current, whereas the plasmonic OPD exhibits a significantly suppressed dark current. It indicates that the introduction of Ag NPs changes the contact on the cathode side. Here, the control OPD exhibits the unipolar conductivity property in the dark and it works as an open circuit under reverse bias, from which one can deduce that on the ITO anode side, the Schottky contact is formed, and on the Al cathode side, the Ohmic contact is formed. Here, the Ohmic contact at the Alq_3_/Al interface is similar to that which occurred in Ref. [42]. It is also deduced that after incorporating Ag NPs, the Ohmic contact on the Al side changes into the Schottky type, responsible for the significantly reduced dark current above 2 V. The energy diagrams in the dark are then displayed in Appendix A for further analysis. Overall, the significantly suppressed dark current due to the formed Schottky junction, on the premise that the amount of free carriers in Alq_3_ generated by the PHCT effect under illumination can surpass that of background carriers under a non-zero bias.

Thick curves with symbols in Figure 2a,b represent the log I−V characteristics of the control OPD and the plasmonic OPD under illumination at different wavelengths of 375 nm (rectangles), 565 nm (circles), 660 nm (diamonds), and 850 nm (triangles), with the incident power density of 13 mW/cm^2^. One can see clearly in Figure 2a that, for the control OPD, the photocurrents under biases above 1.8 V at the wavelengths of 660 nm and 850 nm almost coincide with its dark current. The observations are consistent with the transient photocurrent responses. As shown by the thin curves in Figure 2c,d, the control OPD biased at 2.5 V cannot output any electric signals under pulse illuminations of either 660 nm or 850 nm light. However, for the plasmonic OPD, the photocurrents under biases above 1.8 V are quite distinct from their dark currents at the wavelengths of 660 nm and 850 nm, as observed in Figure 2b. The transient photocurrent responses of the plasmonic OPD under pulse illumination (thick curves in Figure 2c,d) further confirm that, with the incorporation of Ag NPs, the OPD can effectively detect photos at 660 nm and 850 nm, which lie beyond the Alq_3_ intrinsic absorption band. The photo-to-dark current ratio at 2.5 V is calculated to be 479 at 660 nm, corresponding to a responsivity of 2.01 × 10^−6^ A/W. The photo-to-dark current ratio is calculated to be 67 at 850 nm, corresponding to a responsivity of 0.28 × 10^−6^ A/W. The linear I−V plots under illumination at 660 nm and 850 nm are also presented in Appendix A. From these, it is evident that the plasmonic OPD starts conducting at 1.8 V, which means that a 1.8 V or higher forward bias can counterbalance the Schottky junction formed on the ITO anode side, guaranteeing the free carriers in Alq_3_ generated by the PHCT effect can be smoothly collected by the anode.

The transient photocurrent responses at the wavelengths of 375 nm and 565 nm are also characterized as shown in Appendix A. Negligible differences are observed between the responses of the two OPDs at the wavelength of 375 nm. It is observed that the control OPD can also sense light at 565 nm, with a photo-to-dark current ratio of 89 at 2.5 V. This should come from the weak sub-bandgap absorption of Alq_3_. With the assistance of Ag NPs, the photo-to-dark current ratio at 2.5 V increases by a factor of around 34, reaching 3024. Such a large enhancement factor of photo-to-dark current ratio at the wavelength of 565 nm is the product of the suppressed dark current ascribed to the formed Schottky junction, along with the enhanced photocurrent due to the PHCT effect.

In the following, we carry out systematic investigations of surface morphologies and absorptions to verify the enhanced photocurrent from the PHCT effect beyond the Alq_3_ intrinsic absorption band. Figure 3a–c show the SEM images of the multilayer films of ITO/Alq_3_ and ITO/Alq_3_/AgNP, and the plasmonic OPD (ITO/Alq_3_/AgNP/Al), respectively. From Figure 3a, it is seen that the deposited Alq_3_ film exhibits clear grain boundaries, reflecting its polycrystalline property. Because of poor wettability, Ag films have very low adhesion energy at the Ag/Alq_3_ interface. This can also be explained by the fact that Ag atoms interact more strongly with each other than with the substrate [43]; the low-evaporating-rate (0.2 Ås^−1^) deposition of Ag on top of Alq_3_ prefers to form irregular NPs instead of continuous films, as displayed in Figure 3b. EDS maps of Ag NPs are given in Appendix A. This phenomenon of the island-like Ag cluster formation has been investigated in the fabrication of transparent metal top electrodes for photoelectric device both in our previous work [44] and in work conducted by other researchers [45,46,47]. This provided theoretical and experimental evidence for Ag NPs formation on the Aql_3_ layer. Figure 3c has a similar distribution to Figure 3b, indicating that the subsequent process of Al evaporation produces a conformal thin film on top of the irregular Ag NPs. It is noted that the obtained Al thin film is continuous so that the top cathode is effective on the photon-generated carrier collection.

The hybrid AgNP/Al nanostructure is expected to excite rich plasmonic resonances, enhancing the absorption of light compared with the pure Al thin film. The absorption spectrum of the ITO/Alq_3_ structure was firstly taken to confirm that Alq_3_ can only absorb light with the wavelength shorter than ~460 nm through the intrinsic absorption process, as displayed in Figure 3d (triangle curve). After coating an Al layer on ITO/Alq_3_ film, the absorption of light, as represented by the dotted curve in Figure 3d, is enhanced significantly at the broadband wavelength range between 300 nm and 900 nm. It is emphasized that, although Alq_3_ can induce the sub-bandgap absorption process responsible for the photoelectric response displayed by the thin curve in Appendix A, the absorption at wavelength longer than 460 nm from Alq_3_ is pretty weak. It is straightforward to understand that the absorption of light with the wavelength longer than 460 nm must come from the Al layer. Through the incorporation of an additional Ag NP layer, the absorption of the OPD device is further enhanced over broadband range, as shown by the rectangle curve in Figure 3d. The average enhancement factor of absorption, evaluated at the wavelength range between 300 nm and 900 nm, produced by the plasmonic OPD with respect to the control OPD is 1.23.

For the plasmonic OPD, the observed phenomenon of enhanced absorption can be attributed to the excitation of plasmonic resonances. Electromagnetic simulations of the investigated film and devices are carried out to confirm this attribution. Details of the simulation models are described in Appendix A. Here, the irregular Ag NPs are simplified as an assembly of semi-ellipsoids with different sizes. Figure 4a shows the calculated absorption spectra of the ITO/Alq_3_ structure, the control OPD, and the plasmonic OPD, respectively. It is seen in Figure 4a that the calculated absorption spectrum of the control OPD (dotted curve) is in good agreement with the measured spectrum shown in Figure 3d, confirming that the 10 nm-thick Al film plays the role of absorbing broadband light beyond the Alq_3_ intrinsic absorption band. Figure 4a also reflects that the absorption efficiency of the plasmonic OPD is greater than that of the control over the whole investigated range. Three absorption peaks are witnessed, which are at the wavelengths of 430 nm, 540 nm, and 660 nm. Although the curved shapes of the measured and simulation spectra display some differences in peak wavelengths and amplitudes (because the NPs utilized in simulation deviate from the real NPs in geometrical parameters), the evidence is sufficient to support the conclusion that introducing Ag NPs into the Alq_3_-based OPD can significantly improve light absorption over a broadband wavelength range. Here, it is mentioned that, in the experiment, the photoelectric responses were measured using the available LEDs as representatives, which cover the Alq_3_ intrinsic absorption band (375 nm), the Alq_3_ sub-bandgap absorption band (565 nm), and the bands where Alq_3_ cannot absorb light (660 nm and 850 nm).

Figure 4b displays the distributions of normalized |E| of the plasmonic OPD under the three simulation peak wavelengths of 430 nm, 540 nm, and 660 nm, as well as a near-infrared wavelength of 850 nm. The cross-section plane is intentionally selected to clearly show the enhanced localized fields excited by the hybrid AgNP/Al nanostructure. It is apparent that various plasmonic resonances are induced at distinct wavelengths by the hybrid AgNP/Al nanostructure. At all wavelengths, hot spots emerge in the valleys between adjacent NPs. It is deduced that the hybridization of hot spots in neighboring valleys causes the plasmonic resonances to have rich forms. In the experiment, because the irregular NPs differ from each other in shapes and distances, the measured absorption spectrum of the plasmonic OPD has a much smoother profile than the simulation one. Overall, the strong light absorption taking place in the metals of Ag and Al, which are adjacent to the Alq_3_ layer, can generate a large number of hot electrons, being expected to produce considerable photoelectric signals beyond the Alq_3_-intrinsic absorption band.

Finally, the analyses on the charge-transfer processes in the plasmonic OPD under illumination described by the schematic diagrams of energy levels are carried out. Under reverse bias, the hot electrons in Ag and Al produced by plasmonic resonances are attracted by the positive potential applied on the cathode, resulting in no carrier transfer into the Alq_3_ layer, and thereby the photoelectric signal cannot be generated at long wavelengths. In the situation of forward bias below 1.8 V, as shown in Figure 5a, the hot electrons have the possibility to transfer into the Alq_3_ layer due to the repulsion from the electric power, but the transfer of electrons from Alq_3_ into the ITO anode is still inhibited due to the remaining barrier on the anode side. When the forward bias is sufficiently high, being above 1.8 V (as derived from Appendix A), the Schottky barrier on the ITO anode side can be fully counterbalanced by the applied voltage, resulting in the smooth flow of carriers from Alq_3_ into the ITO anode, as shown in Figure 5b. This way, the hot carriers excited by the plasmonic resonances at long wavelengths can eventually produce electric signals in the external circuit. The effect of forward bias on the Schottky junction on the cathode side is to broaden the barrier, facilitating the flow of electrons in Alq_3_ by the drift process. Here, it is mentioned that in the control OPD, although the Al cathode can absorb some light at long wavelengths, its Ohmic contact feature prevents the hot electrons from being distinguished from the injected electrons under forward bias. This is the reason for the coincidence of dark I−V and light I−V at 660 nm and 850 nm for the control OPD, as shown in Figure 2a.

## 4. Conclusions

In summary, we have demonstrated that, with the help of noble metal nanostructures, an OPD can produce sensitivity beyond the intrinsic absorption band produced by organic semiconductors. The sensitivity was realized through the generation of plasmonic hot carriers in metals, which could then transfer into organic semiconductors through the PHCT process, consequently inducing the photoelectric signals. The low-cost material Alq_3_ was selected as an example to accept the transferred electrons. It has been found that introducing Ag NPs into the ITO/Alq_3_/Al OPD can induce rich plasmonic resonances on one hand, and on the other, it can suppress the dark current by forming a Schottky junction. As a result, the plasmonic OPD can output apparent photoelectric signals in the wavelength range beyond Alq_3_ intrinsic absorption wavelength edge (460 nm). At an infrared wavelength of 850 nm, the proposed plasmonic OPD can stably produce responses by pulsed illumination. In contrast, the ITO/Alq_3_/Al OPD failed to output any signals under 850 nm illumination. It is emphasized that the general validity of PHCT-enhanced photodetection performances in organic devices is noted because they possess merits, including being low in cost, mechanically flexible, and are capable of large-area fabrication. As long as infrared OPDs are successfully realized when resorting to the PHCT process, the cost of infrared photodetectors can experience a cliff drop, which could promote their applications in areas of wearable bio-sensing electronics, industry vision, and so forth.

## Figures and Tables

**Figure 1 nanomaterials-12-03084-f001:**
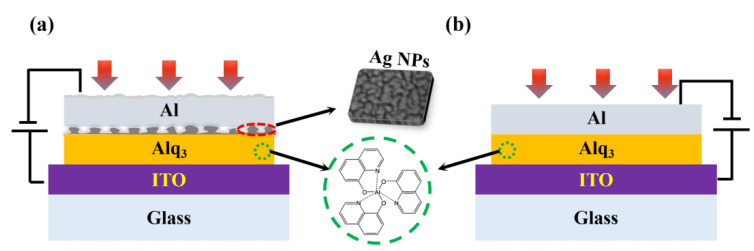
(**a**,**b**) Schematic diagrams of OPDs in cross-sectional view composed of ITO/Alq_3_/AgNP/Al and ITO/Alq_3_/Al. The insets in the middle are the molecular structure of Alq_3_ (**bottom**) and the schematic diagram of the Ag NP layer (**top**).

**Figure 2 nanomaterials-12-03084-f002:**
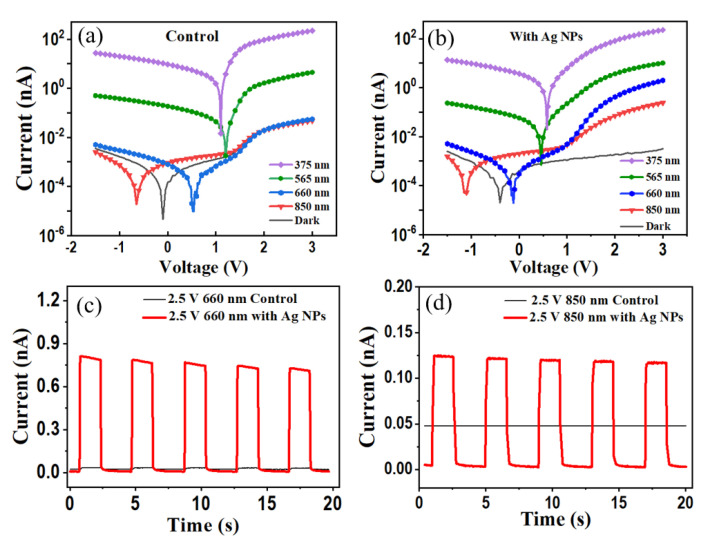
(**a**,**b**) Logarithmic I−V characteristics of the control OPD and the plasmonic OPD in the dark and under illumination at 375 nm, 565 nm, 660 nm, and 850 nm. (**c**,**d**) Transient current responses of different OPDs using 660 nm and 850 nm pulse LEDs as illumination sources.

**Figure 3 nanomaterials-12-03084-f003:**
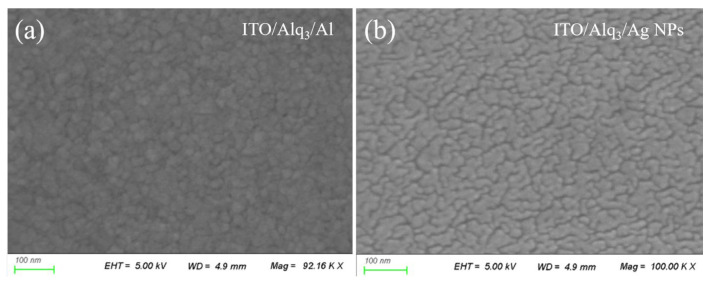
(**a**–**c**) SEM images of top surfaces of the ITO/Alq_3_ film, the ITO/Alq_3_/AgNP film, and the ITO/Alq_3_/AgNP/Al OPD. (**d**) Measured absorption spectra of the ITO/Alq_3_ structure, the control OPD (ITO/Alq_3_/Al), and the plasmonic OPD (ITO/Alq_3_/AgNP/Al).

**Figure 4 nanomaterials-12-03084-f004:**
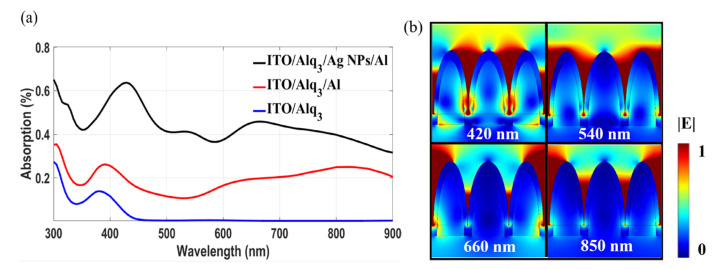
(**a**) Calculated absorption spectra of the ITO/Alq_3_ film, the control OPD (ITO/Alq_3_/Al), and the plasmonic OPD (ITO/Alq_3_/AgNP/Al). (**b**) Maps of |E| for plasmonic OPD at the wavelengths of 420 nm, 540 nm, 660 nm, and 850 nm.

**Figure 5 nanomaterials-12-03084-f005:**
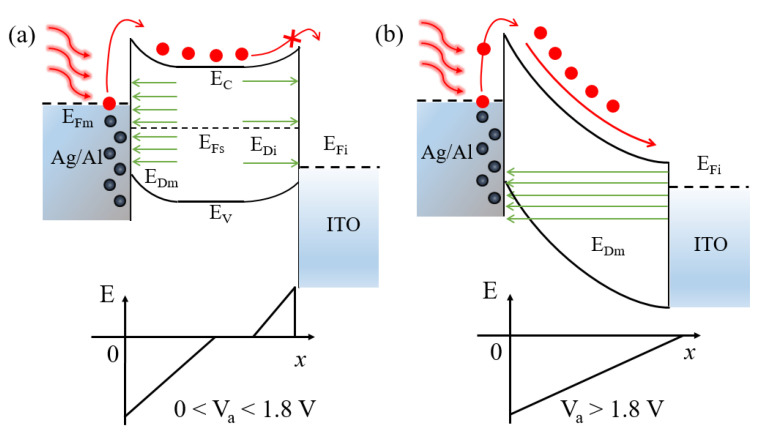
Schematic energy level diagrams of the plasmonic OPD under forward bias below 1.8 V (**a**) and above 1.8 V (**b**), when the illumination is on at the wavelength beyond the Alq_3_ absorption band.

## Data Availability

Not applicable.

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
