# Peer review of "Organic Photodetectors with Extended Spectral Response Range Assisted by Plasmonic Hot-Electron Injection"

_nanomaterials, 2022, doi:10.3390/nano12173084_

Round 1

Reviewer 1 Report

NP decorations to improve the spectral bandwidths of OPDs through plasmonic resonance is a recently emerging technology. The authors incorporated AgNPs into classical control OPD devices and reported plasmonic resonance enhanced photo-detection. The results are well organized and described. It would be a scientific interest to the related fields. However, the authors missed several crucial data. I would recommend this report for further consideration. Before that, the authors should provide a major revision to the manuscript.

My suggestions to the authors: 

Q1. I have two concerns about Fig. 1(a). The first is, Why there is a space between AgNP and Al in your schematic? why AgNPs are scattered out? It’s really confusing, in my opinion. If I am mistaken, please describe it in detail. Otherwise, I suggest authors to reproduce Fig.1(a).

My other concern is that a more attractive design schematic can be produced. It is just my recommendation to the authors and doesn’t affect the article's assessment.

Q2. Fig. 3, did you perform the elemental analysis characterizations? If yes then, Could you include it in the supplementary?

Somehow, I couldn’t see AgNPs in the SEM images. Any of the microstructural studies that reflect the AgNP layer formation would be suggested.

Q3. Is this device optimized in terms of AgNP layer thickness and AgNP size?

Q4. Would you present a schematic for plasmonic resonance oscillations of AgNP?

The novelty of the device will be very clear if the authors provide a schematic of the underlying mechanism for effective photo-detection for all the wavelengths (at least for a few wavelengths) and a possible comparison between control and plasmonic OPDs.

Q5. Authors may compare the obtained results with other NPs, for instant Au.

Overall, I would recommend this paper for further consideration. Anyway, a major revision must be done before being considered for publication in Nanomaterials.

Reviewer 2 Report

This article presents organic photodetector based on Alq3, and enhancements possible by adding Ag nanoparticles, which include higher sensitivity and lower noise. Sample preparation and measurement methods are well described, and the conclusions are supported by results presented. The article should be of interest to the field of organic photodetector development. It is well prepared, with only the following minor comments suggested:

line:comment

76: abbreviation ITO not defined
149: Ref.[38] . (extra space)
Figure 4.: periodic boundary conditions? If periodic boundary was applied, I would expect the result to be the same for every period of NP structure. It is apparently not so, the middle NP structure is surrounded by different |E| response as it's two neighbors, with strongly different response at image edges. How is this so?
232: is in good consistent -> is consistent/is in good agreement
244: and the band Alq 3 cannot absorb light -> and the bands where ...
250: It is observed apparently -> It is apparent
252: the plasmonic resonances have rich forms -> the plasmonic resonances to have rich forms
281: , but its Ohmic contact feature makes the hot electrons cannot be distinguished -> , its Ohmic contact feature prevents the hot electrons to be distinguished
291: OPD, can -> OPD can

Round 2

Reviewer 1 Report

The authors considered my suggestions and revised accordingly. I would recommend publishing in the present form.